# A Longitudinal Decline in Walking Speed Is Linked with Coexisting Hypertension and Arthritis in Community-Dwelling Older Adults

**DOI:** 10.3390/jcm13185478

**Published:** 2024-09-15

**Authors:** Saud M. Alrawaili, Khalid Alkhathami, Mohammed G. Elsehrawy, Mohammed S. Alghamdi, Hussein M. Alkahtani, Norah A. Alhwoaimel, Aqeel M. Alenazi

**Affiliations:** 1Department of Health and Rehabilitation Sciences, Prince Sattam Bin Abdulaziz University, Al-Kharj 11942, Saudi Arabia; s.alrawaili@psau.edu.sa (S.M.A.); n.alhwoaimel@psau.edu.sa (N.A.A.); 2Department of Health Rehabilitation, Shaqra University, Shaqra 11961, Saudi Arabia; kalkhthami@su.edu.sa; 3Department of Nursing, Prince Sattam Bin Abdulaziz University, Al-Kharj 11942, Saudi Arabia; m.elsehrawy@psau.edu.sa; 4Department of Medical Rehabilitation Sciences, Faculty of Applied Medical Sciences, Umm Al-Qura University, Makkah 21955, Saudi Arabia; msghamdi@uqu.edu.sa; 5Jubail Miltary Hospital, Al Jubail 35517, Saudi Arabia; dr.al-kahtani@hotmail.com

**Keywords:** hypertension, arthritis, gait speed, mobility, elderly

## Abstract

**Objective**: The aim was to investigate the association between baseline coexistence of hypertension (HTN) and arthritis, HTN alone, or arthritis alone, and their impact on longitudinal physical function measures among community-dwelling older adults over 5 years of follow-up. **Methods**: Ours was a longitudinal prospective cohort study from the second wave (2010–2011) and third wave (2015–2016) of the National Social Life, Health, and Aging Project (NSHAP). Data for older adults were used. Participants were categorized based on self-reported diagnoses into four groups: coexisting HTN and arthritis, HTN only, arthritis only, or neither. Physical performance measures included walking speed using the 3-Meter Walk Test and the Five Times Sit-to-Stand Test (FTSST). Multiple generalized estimating equations with linear regression analyses were conducted, adjusting for age, sex, race, body mass index (BMI) educational level, pain severity, and baseline use of pain and hypertension medications. **Results**: Data for 1769 participants were analyzed. Slower walking speed was only associated with coexisting HTN and arthritis (B = −0.43, *p* < 0.001) after accounting for covariates. The coexisting HTN and arthritis group showed no significant association with FTSST (B = 0.80, *p* = 0.072) after accounting for covariates. **Conclusions**: The coexistence of baseline HTN and arthritis in older adults is associated with a gradual decline in only walking speed as a physical performance measure in older adults.

## 1. Introduction

Arthritis is a common degenerative disease that has many forms and affects older adults [1]. The common types of arthritis are osteoarthritis and rheumatoid arthritis, with similar and different manifestations [1]. Osteoarthritis is a chronic degenerative disease affecting joints and characterized by the gradual breakdown of cartilage, leading to symptoms including pain, stiffness, and limitation in mobility and physical performance [2]. In contrast, rheumatoid arthritis is considered an autoimmune disease, leading to joint inflammation and joint symptoms including pain and swelling [2]. Despite their differences, both conditions share some similarities, including their negative impact on daily activities, quality of life, and physical performance in men and women [3,4]. Osteoarthritis and rheumatoid arthritis have been associated with chronic conditions such as metabolic diseases including diabetes and hypertension [5,6].

Hypertension (HTN) and arthritis are common conditions in elderly people, and their coexistence is not uncommon [7,8,9]. Reports indicate that the occurrence of HTN in patients with arthritis varies according to different studies [8,10]. Data from the National Health and Nutrition Examination Survey (1999–2018) for a cohort of 48,372 participants showed that 48% and 24.5% had HTN and arthritis, respectively [8]. Additionally, the authors reported a significant association between HTN and arthritis, after adjusting for demographics, lifestyle, and health factors [8]. In a systematic review and meta-analysis, Lo et al. [10] found that the prevalence of HTN was significantly higher in individuals with knee osteoarthritis (OA) (42.6%) compared to those without it (20.1%). Similar trends were observed in cases of hand OA and generalized OA, where HTN prevalence was markedly higher in OA patients compared to those without OA [10].

Metabolic disorders have shared characteristics and potential mechanisms as systemic diseases affecting all body parts (including joints) and have been linked with arthritis. The coexistence of metabolic disorders, including diabetes and HTN, with arthritis has been explored in the literature, focusing on aspects such as medication usage [11,12] and functional outcomes [13,14,15,16]. In multiple studies, Alenazi and colleagues [13,14,15] found correlation between diabetes and a decline in physical performance in older adults. This decline is particularly evident in reduced gait speed and increased severity of joint pain, especially in individuals suffering from comorbid conditions such as diabetes and OA. In a retrospective study, authors found that diabetes is associated with regional knee pain and significantly increases pain severity while walking in individuals with knee osteoarthritis [15]. These findings emphasize the complex influence of metabolic disorders, specifically diabetes, on pain and mobility in individuals with osteoarthritis.

The relationship between the coexistence of HTN and arthritis with physical function has not been fully explored. Current evidence suggests that HTN is associated with mobility decline in older adults, particularly reduced walking speed [17]. A longitudinal cohort study found that HTN contributes to gait speed decline in older adults over time, irrespective of whether the HTN is newly diagnosed, uncontrolled, or controlled [18]. This association was found to be independent of other health-related factors, including brain, kidney, and heart function [18]. Another study reported that HTN was associated with a reduced walking speed at baseline and a greater annual decline in walking speed in older adults [19]. Additionally, individuals with HTN had significantly higher gait variability compared to normotensive individuals, with notable differences in gait symmetry and speed [20]. Collectively, these findings suggest that HTN plays a role in motor decline in older adults.

Decline in physical function in older adults, especially those with comorbidities, is multifactorial and complex [21,22,23]. Preliminary evidence suggests that HTN is associated with decline in physical function, as evidenced by slower gait speed [18,19,20]. Additionally, there is abundant research on the impact of arthritis, both osteoarthritis and rheumatoid arthritis, on physical function [24,25,26]. However, there is a gap in the literature on whether coexistence of HTN and arthritis influences physical function in older adults. Therefore, the primary aim of the study was to investigate the association between baseline coexistence of HTN and arthritis including osteoarthritis or rheumatoid arthritis, HTN alone, or arthritis alone, and their impact on longitudinal physical function measures among community-dwelling older adults over 5 years of follow-up.

## 2. Methods

### 2.1. Study Design and Participants

The current study employed a longitudinal design from wave 2 and wave 3 of the National Social Life, Health, and Aging Project (NSHAP). NSHAP is a longitudinal survey which is nationally representative of older adults in the United States. The main goal for this dataset was to appraise the impact of social, biological, emotional, and environmental factors on health and aging [27]. Data for NSHAP project were gathered in three waves, each a 5-year interval: 2005–2006 for wave 1, 2010–2011 for wave 2, and 2015–2016 for wave 3. However, since we were targeting older adults (age ≥ 60 years) and to minimize missing data on some of the key variables and participation, the wave 2 and wave 3 were selected with data obtained from 2010–2011 and 2015–2016 for the current study. The selection of wave 2 and wave 3 was because the primary outcome variables, walking speed and Five Times Sit-to-Stand results, were not included in the first wave from 2005 until 2006, and some of the key confounders such as pain severity were not included in wave 1. In the NSHAP, data were gathered using three methods: (a) individual interviews at the subjects’ homes; (b) biometric measurements; and (c) leave-behind, self-reported questionnaires with questions and items regarding chronic diseases such as HTN and arthritis and pain severity. Informed consent was given to each participant, and institutional review board approval was obtained.

### 2.2. Outcome Measures and Covariates

#### Groups

Participants were asked about chronic diseases with the question: “Have you been diagnosed with arthritis, HTN, or both?”. Based on their responses, participants were categorized into four distinct groups: (1) HTN only, (2) arthritis only, (3) coexisting HTN and arthritis, and (4) neither HTN nor arthritis.

### 2.3. Walking Speed

Walking speed was measured using a 3-Meter Walk Test adapted from the Short Physical Performance Battery [28,29]. Participants were instructed to walk twice at their usual pace. The use of assistive devices such as canes or walkers was permitted. The durations of both trials were documented in seconds.

### 2.4. The Five Times Sit-to-Stand Test (FTSST)

Functional lower limb strength was assessed using *FTSST* adapted from the Short Physical Performance Battery [28,29]. Participants were initially instructed to execute a single chair stand without using their arms, with the interviewer providing a demonstration. The proper technique involved feet firmly planted on the ground, leaving about one hand’s width between the knee and the chair, arms folded across the chest, and attempting to stand while keeping the arms folded. If the single chair stand was successful, participants were then asked to perform the Five Times Sit-to-Stand Test, which involved completing five consecutive chair stands as rapidly as possible. The time was recorded after the participant fully stood up on the fifth stand. The time taken to complete the Five Times Sit-to-Stand was documented in seconds.

### 2.5. Covariates

Different covariates were considered established risk factors for physical performance among older adults. Therefore, we included demographics (age, sex, race), body mass index (BMI), education, pain severity, pain medications, and hypertension medications in the current data analysis. Age was measured as a continuous variable in years. Sex was dichotomized into male or female. Race was categorized into white, black, Hispanic, and other. BMI was calculated using weight in pounds divided by squared height in inches and multiplied by 703. The education level of the studied groups was categorized as less than high school, high school, associate degree, and bachelor’s degree and higher.

Pain severity was measured over four weeks using the Verbal Descriptor Scale (VDS). This scale has good validity and reliability for quantifying pain severity among older adults [30,31,32]. The VDS has seven different levels of pain intensity, and its scoring ranges from 0 to 6, with higher scores indicating more extreme pain. The VDS for pain severity has shown validity and reliability, specifically for older adults with different racial and cognitive backgrounds [32,33,34].

Pain medications in the current data had one of eight classifications (i.e., central nervous system medications, analgesics, miscellaneous analgesics, narcotic analgesics, non-steroidal anti-inflammatory drugs, analgesic combination, cox2 inhibitors, and salicylates). Each participant was asked about using each type of pain medication and responded with yes/no answers. Afterwards, these medications were combined into one variable named “pain medications”, with yes/no answers for using any of the eight types.

Hypertension medications in the current data included four classifications (i.e., antihypertensive combinations, angiotensin II inhibitors, cardio-selective beta blockers, and non-cardio-selective beta blockers). Each participant was asked specifically about using each type of medication and responded with yes/no answers. Then, these medications were combined into one variable named “hypertension medications”, with yes/no answers for using any of the four types.

### 2.6. Statistical Analysis

Descriptive statistics (mean, standard deviation or counts, percentages) were used to examine demographics and clinical characteristics for all participants. Demographics and clinical variables were compared between groups (i.e., HTN only, arthritis only, coexisting HTN and arthritis, neither HTN nor arthritis) using a one-way ANOVA for continuous variables or Chi-square tests for categorical variables.

Generalized estimating equations (GEEs) with multiple linear regression were utilized to examine the association between groups with walking speed (measured through the 3 m walk test) and functional lower limb strength (measured through the FTSST). Two models were computed, one unadjusted and one adjusted. The adjusted model was controlled for age, sex, race, BMI, educational level, pain severity at baseline and follow-up (wave 2 and wave 3), and baseline pain medications and hypertension medications. Results were presented as unstandardized coefficients (B) with standard error. The alpha level was set at a *p*-value of 0.05, and all the analyses used IBM SPSS for Mac version 25.0 (SPSS Inc., Chicago, IL, USA).

## 3. Results

A total of 3196 older adults at baseline (wave 2) were included in this longitudinal study. However, due to missing data on the key confounding variables on physical performance such as pain severity, the number of participants at baseline was 1791. The age of participants included at baseline (wave 2) ranged from 62 to 91 years. Significant differences were observed at baseline between groups in age, sex, race, BMI, education levels, pain severity in the past 4 weeks, walking speed, and Five Times Sit-to-Stand Test time. Furthermore, walking speed was significantly slower in the coexisting HTN and arthritis group (mean = 0.61 m/s) and faster in the neither HTN nor arthritis group (mean = 0.74 m/s). Similarly, the time required to complete the FTSST was significantly slower in the coexisting HTN and arthritis group (mean = 15.60 s) and faster in the neither HTN nor arthritis group (mean = 13.39 s). Table 1 shows demographic and clinical variables across groups at baseline (wave 2)

Table 2 shows the results of the GEE with a linear regression model examining the association between groups and walking speed over time, with unadjusted and adjusted models. In the model adjusted for age, sex, race, BMI, education, baseline use of pain and hypertension medications, and pain severity at baseline and follow-up (wave 2 and wave 3), only the coexisting HTN and arthritis group showed significant association with walking speed (B = −0.043, *p* < 0.001). However, other groups, including the arthritis only and HTN only group, showed no significant associations with walking speed.

Table 3 shows the results of the GEE with a linear regression model examining the association between groups and FTSST over time with unadjusted and adjusted models. In the unadjusted model, all groups showed significant associations with FTSST, including coexisting HTN and arthritis (B = 2.05, *p* < 0.001). However, in the model adjusted for age, sex, race, BMI, education, baseline use of pain and hypertension medications, and pain severity at baseline and follow-up (wave 2 and wave 3), there was no significant association between all groups, including coexisting HTN and arthritis with FTSST (B = 0.80, *p* = 0.072).

## 4. Discussion

This study aimed to explore the association between the coexistence of HTN and arthritis and their impact on physical function among community-dwelling older adults. The findings revealed a significant decline in walking speed over time only for those with coexisting HTN and arthritis in the fully adjusted model. Other groups showed no significant association with walking speed. The model adjusted for age, sex, race, BMI, education, baseline use of pain and hypertension medications, and pain severity at baseline and follow-up showed no significant association between coexisting HTN and arthritis with functional lower limb strength measured by FTSST. This might indicate a mediating effect of covariates such as BMI and medications.

This research aligns with prior studies that have revealed that both HTN and arthritis have a significant negative impact on physical function, especially walking speed, among older adults. Notably, this study adds to the body of evidence by illustrating the compounded impact of coexisting HTN and arthritis on longitudinal changes in physical performance measures. Previous studies have found a high prevalence of HTN in individuals with knee osteoarthritis, suggesting a possible link between joint degeneration and systemic vascular conditions [10]. Our study expands upon that research by demonstrating that the presence of both conditions exacerbates functional decline more than either condition alone. The relationship between metabolic disorders and physical function decline is well established, as demonstrated in studies by Alenazi et al. [13,14,15], which showed an association between diabetes and a decline in physical performance. These studies are consistent with our findings on HTN and suggest a broader pattern of metabolic conditions that contribute to reduced physical functions.

In terms of mobility, previous research has shown that HTN is associated with reduced walking speed and increased gait variability [17,18,19,20]. This study extends those findings by demonstrating that the combination of HTN and arthritis leads to an even greater decline in walking speed and functional lower limbs strength than HTN alone. These results emphasize the intricate nature of physical function decline in older adults, as previously described [21,22,23]. This implies that interventions targeting multiple risk factors may be necessary to maintain physical functions in older adults. Although some studies in the literature address the influence of HTN and arthritis on physical functions separately, there is still a gap regarding their combined effect. This study contributes to filling this gap by demonstrating that the coexistence of these conditions is associated with a more significant decline in physical functions such as walking speed than either condition alone.

Our finding that coexisting arthritis and HTN were not significantly associated with a decline in functional lower limb strength over time in older adults contradicts the findings of Luo et al. [35]. In their study, they observed that relative muscle strength (RMS) was negatively correlated with HTN. This suggests that higher muscle strength is an independent protective factor against HTN. However, the mechanisms underlying this association were not fully clear, though they suggested it might relate to myosteatosis-induced insulin resistance, which is closely associated with the risk of developing HTN.

On the other hand, Blanchard et al. [36] found a contrasting result, where muscle strength was positively associated with HTN. They hypothesized that this might be due to shifts in muscle fiber type from Type I to Type IIb/x and a transition from oxidative to glycolytic metabolism in individuals with high blood pressure. This shift in muscle fiber type could result in increased muscle strength in hypertensive individuals compared to those with normal blood pressure. The discrepancy between studies might be explained by differences in the physiological adaptations in muscle fiber types and metabolism in hypertensive individuals.

Findings on the coexistence of HTN and arthritis among community-dwelling older adults have several important clinical implications. First, the complexity of health issues in older adults, especially those with coexisting conditions, requires a comprehensive multidisciplinary approach among healthcare providers. The study highlights the importance of regular assessments of physical functioning, including walking speed, as part of routine assessments and follow-ups. Lifestyle interventions, such as physical fitness exercises, are recommended to improve and maintain optimal physical functions. Finally, careful medication management for both HTN and arthritis is important, considering potential drug interactions and the overall health of the individual.

Although this study utilized a longitudinal design over 5 years with two waves of data for the same participants, it has some limitations. Self-reported rates of HTN and arthritis may underestimate their prevalence, given participants may not have been diagnosed with these conditions. Another limitation is recall bias, especially in older adults when asked about their history of diseases. A high percentage of the included older adults used pain medication in the current study, ranging from 52% in the control group to 74% in the coexisting HTN and arthritis group. However, the use of pain medication may not affect results, even after controlling for this covariate, as the use of medications including pain medications was not associated with increased risk of fall among older adults in previous research [37,38]. Another limitation that should be mentioned is that the classification of hypertension medications included some medications that have multiple uses, such as angiotensin II inhibitors, that can be used for HTN and other chronic conditions such as heart failure and kidney disease. Therefore, some participants without HTN used hypertension medications in the current study. Moreover, some participants used steroids (glucocorticoids) at baseline in wave 2, which could be used for rheumatoid arthritis. However, only 66 participants used them, and the results remained similar when adding glucocorticoids as a covariate, although they were not reported. Furthermore, specific types of arthritis, such as osteoarthritis and rheumatoid arthritis, were not captured in this database. The joints affected by arthritis were not specified by the participants, which may limit the generalizability of the results. Future research should examine arthritis and its locations, including weight-bearing and non-weight bearing joints (such as knees and hands), using objective methods of assessment such as radiographs and symptoms, as wel as specifying the types and affected joints.

## 5. Conclusions

Our study’s findings highlight the significance of acknowledging the joint influence of high blood pressure and arthritis on decline in the physical function of older community-dwelling adults. Coexisting HTN and arthritis were associated with a decline in walking speed only as a physical performance measure. These findings highlight the necessity of adopting a multidisciplinary management strategy that takes into account coexisting chronic conditions. Clinical interventions and public health measures should be designed to address the multifactorial nature of physical decline in the aging population, considering the combined impact of high blood pressure and arthritis.

## Figures and Tables

**Table 1 jcm-13-05478-t001:** Baseline demographics characteristics and clinical variables at baseline (wave 2).

Variables	Coexisting HTN and Arthritis n = 499	Arthritis Only n = 249	HTN Only n = 617	Neither HTN nor Arthritis n = 404	*p*-Value
Age, years (mean ± SD)	73.90 ± 7	73.94 ± 7	72.47 ± 7	71.66 ± 7	<0.001
Sex, female, n (%) within groups	210 (64.8)	111 (62.7)	196 (45.9)	141 (45.9)	<0.001
BMI, kg/m^2^ (mean ± SD)	31.49 ± 6.9	28.50 ± 5.7	29.95 ± 6.1	28.24 ± 6.3	<0.001
Race, n (%) within groups					<0.001
White	220 (68.1)	136 (77.7)	305 (71.8)	256 (83.4)	
Black	63 (19.5)	16 (9.1)	63 (14.8)	26 (8.5)	
Hispanic	36 (11.1)	17 (9.7)	42 (9.9)	23 (7.5)	
Others	4 (1.2)	6 (3.4)	15 (3.5)	2 (0.7)	
Education, n (%) within groups					<0.001
Less than high school	86 (26.5)	27 (15.3)	74 (26.5)	39 (12.7)	
High school	87 (26.9)	39 (22.0)	109 (25.5)	69 (22.5)	
Some college/associate degree	89 (27.5)	59 (33.3)	142 (33.3)	94 (30.6)	
Bachelor’s or more	62 (19.3)	52 (29.4)	102 (23.9)	105 (34.2)	
Hypertension medications, yes, n (%) within groups	300 (61.0)	50 (20.6)	381 (62.4)	78 (19.6)	<0.001
Pain medications, yes, n (%) within groups	364 (74.0)	172 (70.8)	371 (60.7)	208 (52.4)	<0.001
Pain severity, (mean ± SD)	3.22 ± 1	3.06 ± 1	2.78 ± 1	2.74 ± 1	<0.001
Walking speed, m/s (mean ± SD)	0.61 ± 0.21	0.67 ± 0.22	0.68 ± 0.21	0.74 ± 0.22	<0.001
Five Times Sit-to-Stand Test, s (mean ± SD)	15.60 ± 7.2	14.91 ± 7.0	14.25 ± 5.7	13.39 ± 5.4	<0.001

Note. SD: standard deviation. Boldface *p*-values indicate a statistically significant difference using a one-way ANOVA or Chi-square test.

**Table 2 jcm-13-05478-t002:** Linear regression analyses for the association between baseline coexisting HTN and arthritis groups with walking speed over time.

Groups	Unadjusted Model (n = 2038)			Adjusted Model (n = 1389)		
	B (95% CI)	SE	*p*-Value	B (95% CI)	SE	*p*-Value
Coexisting HTN and Arthritis	−0.13 (−0.16, −0.11)	0.013	<0.001	−0.043 (−0.07, −0.14)	0.01	<0.001
Arthritis only	−0.079 (−0.11, −0.049)	0.015	<0.001	−0.031 (−0.06, 0.001)	0.01	0.061
HTN only	−0.068 (−0.09, −0.045)	0.012	<0.001	−0.023 (−0.05, 0.001)	0.01	0.065
Neither HTN nor Arthritis	Reference			Reference		

Note. n = number of participants; SE = standard error; CI = confidence interval; HTN = hypertension; The adjusted model was controlled for age, sex, race, BMI, education, pain medications, hypertension medications, and pain level in wave 2 (baseline) and wave 3.

**Table 3 jcm-13-05478-t003:** Linear regression analyses for the association over time between baseline coexisting HTN and arthritis groups and Five Times Sit-to-Stand Test.

Groups	Unadjusted Model (n = 1816)			Adjusted Model (n = 1222)		
	B (95% CI)	SE	*p*-Value	B (95% CI)	SE	*p*-Value
Coexisting HTN and Arthritis	2.05 (1.36, 2.74)	0.35	<0.001	0.80 (−0.07, 1.67)	0.44	0.072
Arthritis only	1.28 (0.43, 2.12)	0.42	0.003	0.66 (−0.24, 1.57)	0.46	0.15
HTN only	0.84 (0.29, 1.38)	0.27	0.003	0.18 (−0.48, 0.85)	0.34	0.58
Neither HTN nor Arthritis	Reference			Reference		

Note. n = number of participants; SE = standard error; CI = confidence interval; HTN = hypertension. The adjusted model was controlled for age, sex, race, BMI, education, pain medications, hypertension medications, and pain level at wave 2 (baseline) and wave 3.

## Data Availability

The NSHAP data are publicly available through the National Archive of Computerized Data on Aging (NACDA, https://www.icpsr.umich.edu/icpsrweb/NACDA/, accessed on 12 June 2024).

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
