# Peer review of "A Longitudinal Decline in Walking Speed Is Linked with Coexisting Hypertension and Arthritis in Community-Dwelling Older Adults"

_jcm, 2024, doi:10.3390/jcm13185478_

Round 1
Reviewer 1 Report
Comments and Suggestions for Authors
Thank you for the opportunity to review your manuscript, “Coexisted Hypertension and Arthritis is Associated with Longitudinal Decline in Physical Function Measures in Community Dwelling Older Adults”
The study aimed to investigate the association between baseline coexistence of hypertension and arthritis, hypertension alone, or arthritis alone, and their impact on longitudinal physical function measures among community-dwelling older adults over 5 years of follow-up.
The study is well laid out and has an appropriate and coherent methodology. The writing is clear and easy to follow, highlighting the essential aspects. The authors present the results and identify the limitations of the study. The conclusions are in line with the study without overestimating the findings.
I have found only one minor typographical error in Table 1, which I recommend the authors correct.
I recommend reviewing the citation system, which is not the one proposed by the journal.
Author Response
Dear Editor and reviewers,
Thank you for your valuable comments that helped us in improving our manuscript. We responded to each comment and highlighted the changes in boldface across the manuscript. It should be noted that after the addition of the BMI as suggested by one of the reviewers, the results have changed and the coexisted HTN and arthritis were no longer significantly associated with 5 times sit to stand measure. We updated the results and tables and other parts of the manuscript. You will find point by point response to comments below.
Reviewer 1
Thank you for the opportunity to review your manuscript, “Coexisted Hypertension and Arthritis is Associated with Longitudinal Decline in Physical Function Measures in Community Dwelling Older Adults”
The study aimed to investigate the association between baseline coexistence of hypertension and arthritis, hypertension alone, or arthritis alone, and their impact on longitudinal physical function measures among community-dwelling older adults over 5 years of follow-up.
The study is well laid out and has an appropriate and coherent methodology. The writing is clear and easy to follow, highlighting the essential aspects. The authors present the results and identify the limitations of the study. The conclusions are in line with the study without overestimating the findings.
Comment: I have found only one minor typographical error in Table 1, which I recommend the authors correct.
RESPONSE: Corrected
Comment: I recommend reviewing the citation system, which is not the one proposed by the journal.
RESPONSE: the citation was revised accordingly.
Reviewer 2
Thank you for the opportunity to read and review your manuscript.
The association of arterial hypertension, arthritis, and their coexistence in the elderly population is well-known. Both these factors and their combination altered physical activity and quality of life.
The study is well organized, and the manuscript is well written, but the results are predictable and are not new. The actuality is presented in the manuscript.
The tests, used in the study are simple and reproducible, the studied population is big, the results are clear, and the discussion contains contemporary relevant literature
The conclusion supports the results
During reading the manuscript I had several questions
Comment: 1) Can you provide more information about the arthritis? Is it only osteoarthritis or rheumatoid arthritis and ankylosing spondylitis? If patients had inflammatory arthritis, please provide the data about the treatment and disease duration. Especially questions are about systemic corticosteroids.
RESPONSE: In this study, arthritis included osteoarthritis and/or rheumatoid arthritis. We provided further information to clarify the types of arthritis in the introduction and methods. With regards to the systemic corticosteroids, glucocorticoids were administered at baseline for some participants and only 66 participants used this medication. When we run the analysis again after the addition of glucocorticoids, the results remain the same. We reported this in the limitation section without reporting for the results since the main results were similar with and without the addition of glucocorticoids as a covariate.
Comment: 2. Also please provide the main anatomical localization of the arthritis, e.g. hip or knee, because they severely influence the patient's mobility
RESPONSE: Thank you for your comment. Unfortunately, there is no information about the arthritis locations such as lower extremity including knee and hip joints. This was already included in the limitation section and future direction in the discussion section.
Comment: 3. Why do patients in the control group receive so many analgetics? What was the reason and how it can influence the study results, e.g. back pain, etc
RESPONSE: Thank you for your comment. It is expected to have participants with pain and pain medications especially among older adults. We discussed this in the limitation section in the discussion section as pain medications were not associated with physical performance and risk of fall in older adults.
Comment: 4. Please provide the main comorbidity, e.g. smoking, cerebrovascular diseases. cardiovascular diseases, obesity, diabetes, etc
RESPONSE: Thank you for your comment and suggestion. Adding comorbidities to the covariates will lead to multicollinearity in the current study since some of the conditions will be counted again in the group HTN and arthritis and this affects the results.
Comment: 5. Why does every fifth patient without hypertension receive medications against high blood pressure? It is very strange
RESPONSE: Thank you for your comment. The classification of hypertension medications included some medications that have multiple uses such as angiotensin II inhibitors that can be used for hypertension and other chronic conditions such as heart failure and kidney disease. We included this in the limitation section in the discussion part.
Comment: 6. Please update the limitations, according to the abovementioned questions
RESPONSE: Thank you for your comment. We updated the limitation accordingly.
Reviewer 3
The study is interesting and well-written. I have only minor comments.
In the abstract authors should clarify that they include both rheumatoid arthritis and osteoarthritis.
Comment: Abbreviations should be consistently used throughout the manuscript. Please check the use of HTN.
RESPONSE: Thank you for your comment. We revised accordingly and consistently.
Comment: It should be clarified at the beginning of the introduction that authors include both rheumatoid arthritis and osteoarthritis.
RESPONSE: We added this to the introduction and aims
Comment: A brief introduction on rheumatoid arthritis and osteoarthritis should be added in the introduction.
RESPONSE: we added a paragraph related to osteo and rheumatoid arthritis in the first paragraph of the introduction section.
Comment: Lines 48-57: this part is unclear. What is the link between diabetes and the manuscript?
RESPONSE: thank you for your comment. Diabetes was mentioned because of its nature as a metabolic disease and systemic disease similar to HTN as a systemic and metabolic disorder and there were studies examined the link extensively with diabetes and arthritis with physical measures. We added a clarification to this paragraph in the introduction section.
Comment: Authors declared that this study is a longitudinal study. However, data were collected and analyzed only at the baseline. There is no follow-up. Am I right?
RESPONSE: Thank you for your comment. This study is a longitudinal design including two waves one at baseline in 2010-2011 and the second wave was collected in 2015-2016. This information was already mentioned in the methods section.
Comment: Could the authors add BMI in table 1?
RESPONSE: Thank you for your comment. Yes, we added BMI to table 1 as suggested and in the methods section and results. We also added the BMI as a covariate for the main analyses and the results have changed. We updated the tables and results, methods, and abstract accordingly.
Comment: Line 189: GEE was not defined.
RESPONSE: Thank you for your comment. We revised accordingly
Comment: It would be important to present all the models applied in the results. I mean authors should report the model without adjusting for age, sex etc and the adjusted model.
RESPONSE: Thank you for your comment. We added the unadjusted model to table 2 and table 3.
Reviewer 2 Report
Comments and Suggestions for Authors
Dear Authors!
Thank you for the opportunity to read and review your manuscript.
The association of arterial hypertension, arthritis, and their coexistence in the elderly population is well-known. Both these factors and their combination altered physical activity and quality of life.
The study is well organized and the manuscript is well written but the results are predictable and are not new. The actuality is presented in the manuscro[t
The tests, used in the study are simple and reproducible, the studied population is big, the results are clear and the discussion contains contemporary relevant literature
The conclusion supports the results
During reading the manuscript I had several questions
1) Can you provide more information about the arthritis? Is it only osteoarthritis or rheumatoid arthritis and ankylosing spondylitis?
If patients had inflammatory arthritis please provide the data about the treatment and disease duration. Especially questions are about systemic corticosteroids.
2. Also please provide the main anatomical localization of the arthritis, e.g. hip or knee, because they severely influence the patient's mobility
3. Why do patients in the control group receive so many analgetics? What was the reason and how it can influence the study results, e.g. back pain, etc
4. Please provide the main comorbidity, e.f. smoking, cerebrovascular diseases. cardiovascular diseases, obesity, diabetes, etc
5. Why does every fifth patient without hypertension receive medications against high blood pressure? It is very strange
6. Please update the limitations, according the abovementioned questions
Author Response

(The authors gave the same response as above.)

Reviewer 3 Report
Comments and Suggestions for Authors
The study is interesting and well-written. I have only minor comments.
In the abstract authors should clarify that they include both rheumatoid arthritis and osteoarthritis.
Abbreviations should be consistently used throughout the manuscript. Please check the use of HTN.
It should be clarified at the beginning of the introduction that authors include both rheumatoid arthritis and osteoarthritis.
A brief introduction on rheumatoid arthritis and osteoarthritis should be added in the introduction.
Lines 48-57: this part is unclear. What is the link between diabetes and the manuscript?
Authors declared that this study is a longitudinal study. However, data were collected and analyzed only at the baseline. There is no follow-up. Am I right?
Could the authors add BMI in table 1?
Line 189: GEE was not defined.
It would be important to present all the models applied in the results. I mean authors should report the model without adjusting for age, sex etc and the adjusted model.
Author Response

(The authors gave the same response as above.)

Round 2
Reviewer 2 Report
Comments and Suggestions for Authors
Dear Authors!
Thank you for the revised version of the manuscript.
I have no additional comments
Author Response
Thank you, we added CIs for table 2 and 3